# An approach to sulfate geoengineering with surface emissions of carbonyl sulfide

Ilaria Quaglia[1], Daniele Visioni[2], Giovanni Pitari[1], and Ben Kravitz[3,4]

[1]Department of Physical and Chemical Sciences, Università dell'Aquila, 67100 L'Aquila, Italy
[2]Sibley School for Mechanical and Aerospace Engineering, Cornell University, Ithaca, NY 14853, USA
[3]Department of Earth and Atmospheric Science, Indiana University, Bloomington, IN, USA
[4]Atmospheric Sciences and Global Change Division, Pacific Northwest National Laboratory, Richland, WA, USA

**Correspondence:** Ilaria Quaglia (ilaria.quaglia@aquila.infn.it)

**Abstract.** Sulfate geoengineering (SG) methods based on lower stratospheric tropical injection of sulfur dioxide ($SO_2$) have been widely discussed in recent years, focusing on the direct and indirect effects they would have on the climate system. Here a potential alternative method is discussed, where sulfur emissions are located at the surface or in the troposphere in the form of carbonyl sulfide (COS) gas. Two time-dependent chemistry-climate model experiments are designed from year 2021 to 2055, assuming a 40 Tg-S/yr artificial global flux of COS, geographically distributed following the present day anthropogenic COS surface emissions (SG-COS-SRF), or a 6 Tg-S/yr injection of COS in the tropical upper troposphere (SG-COS-TTL). The budget of COS and sulfur species is discussed, as well as the effects of both SG-COS strategies on the stratospheric sulfate aerosol optical depth ($\sim \Delta\tau$=0.080 in years 2046-2055), aerosol effective radius (0.46 $\mu m$), surface $SO_x$ deposition (+8.9 % for SG-COS-SRF, +3.3 % for SG-COS-TTL) and tropopause radiative forcing (RF) ($\sim$ -1.5 W/m$^2$ in All Sky conditions in both SG-COS experiments). Indirect effects on ozone, methane and stratospheric water vapor are also considered, along with the COS direct contribution. According to our model results, the resulting net RF is -1.3 W/m$^2$ for SG-COS-SRF and -1.5 W/m$^2$ for SG-COS-TTL, and it is comparable to the corresponding RF of -1.7 W/m$^2$ obtained with a sustained injection of 4 Tg-S/yr in the tropical lower stratosphere in the form of $SO_2$ (SG-SO2, able to produce a comparable increase of the sulfate aerosol optical depth). Significant changes of the stratospheric ozone response are found in both SG-COS experiments with respect to SG-SO2 ($\sim$5 DU versus +1.4 DU, globally). According to the model results, the resulting UVB perturbation at the surface accounts for -4.3% as a global-annual average (versus -2.4% in the SG-SO2 case), with a springtime Antarctic decrease of -2.7% (versus a +5.8% increase in the SG-SO2 experiment). Overall, we find that an increase in COS emissions may be feasible, and produce a more latitudinally-uniform forcing without the need for the deployment of stratospheric aircrafts. However, our assumption that the rate of COS uptake by soils and plants does not vary with increasing COS concentrations will need to be investigated in future works, and more studies are needed on the prolonged exposure effects to higher COS values in humans and ecosystems.

# 1 Introduction

Reducing part of the incoming solar radiation (known as Solar Radiation Modification, SRM) has been proposed as a strategy for reducing surface temperatures and thus mitigating some of the worst side-effects of the greenhouse gases-induced global warming (Budyko, 1977; Institute of Medicine and National Academy of Sciences and National Academy of Engineering, 1992; Crutzen, 2006). Various methods have been proposed to achieve this: the injection of sulfate precursors into the lower stratosphere to obtain a cloud of aerosols capable of reflective a portion of the incoming sunlight has been, by far, the most studied due to the observation of a similar cooling effect produced by explosive volcanic eruptions in the past (Robock, 2000). While preliminary estimates for the cost of an eventual deployment already exist (Smith and Wagner, 2018), from an engineering perspective there are no known technologies readily available to carry $SO_2$ or any other precursors considered up to now from the ground up to the lower stratosphere in the quantities needed to obtain a noticeable effect on the surface climate (Lockley et al., 2020). Since any proposed compound would quickly react to form sulfate aerosols, they would need to be carried sealed to the desired altitude, and then released, to ensure a high enough lifetime compared to that of the same aerosols in the troposphere (Lamarque et al., 2013).

We explore here a different approach to increasing the aerosol optical depth in the stratosphere, that makes use of emissions of a gaseous precursor of sulfate aerosols: carbonyl sulfide (COS). COS has a long atmospheric lifetime (4 to 6 years; Khalil and Rasmussen, 1984; Ulshofer et al., 1996) due to its very low reactivity in the troposphere. Because of this, it is also uniformly mixed in the atmosphere, with an average concentration of 0.5 ppbv, and therefore it easily reaches the stratosphere: in quiescent volcanic conditions, COS is the main contributor of sulfate aerosols in the Junge layer (Brühl et al., 2012), where after photodissociation by ultraviolet light and oxidation processes, it is turned into $SO_2$ and subsequently oxidized into sulfuric acid, forming sulfate aerosols (Crutzen, 1976). It is naturally produced by various biological processes and environments, such as saline ecosystems, rainwaters (Mu et al., 2004) and biomass burning. Furthermore, it is also produced in various industrial processes (Lee and Brimblecombe, 2016) after $CS_2$ is oxidized. Its chemical life is very long (35 years; Brühl et al., 2012) and thus its main sink is the uptake from oxic soils (Kuhn and Kesselmeier, 2000; Steinbacher et al., 2004) and vegetation (Sandoval-Soto et al., 2005). In the concentrations found in the atmosphere, it is not a toxic gas for humans: negative effects have not been found even at around 50 ppm, which is 100,000 times more than the background mixing ratio, and for long exposure times in mice and rabbits (Svoronos and Bruno, 2002). Higher concentrations than that can, however, be harmful (Bartholomaeus and Haritos, 2006). Not much is known however about the response of ecosystems in the presence of high concentrations of COS: Stimler et al. (2010) showed that high levels of COS enhance the stomatal conductance of some plants, which might in turn have other unforeseen effects; further, Conrad and Meuser (2000) proposed that high COS concentrations may interact with soils and possibly change soil pH. For the reasons listed above, Crutzen (2006) discarded the idea of using surface emissions of COS to increase the stratospheric aerosol burden.

In this work, we use the University of L'Aquila-Climate-Chemistry Model (ULAQ-CCM) to perform simulations to verify if the increase in surface emissions of COS would be a viable form of sulfate geoengineering, by obtaining a stratospheric aerosol optical depth (AOD) similar to that obtained with the injection of 8 Tg-$SO_2$ in the stratosphere. We also perform simulations

where the release of COS is localized in the tropical upper troposphere. This allows us to investigate whether the increase in surface concentrations of COS can be avoided while at the same time circumventing the need to reach altitude currently unattainable with modern aircrafts (Smith et al., 2020). Together with assessing the resulting aerosol cloud, we also explore the eventual side-effects on key chemical components in the atmosphere, in order to determine how the side effects from COS-induced Sulfate Geoengineering compare with those from $SO_2$-induced Sulfate Geoengineering. For the latter, there is ample literature assessing its effect on stratospheric ozone (Tilmes et al., 2008; Pitari et al., 2014; Xia et al., 2017a; Vattioni et al., 2019): the increase in surface area density, stratospheric heating and dynamical effect all play a part in determining the overall changes (Tilmes et al., 2018b; Richter et al., 2017) to the ozone column that, in turn, determine the changes in surface UV (Visioni et al., 2017b; Madronich et al., 2018) that would be important when considering adverse health effects (Eastham et al., 2018).

## 2   Model description and setup of numerical experiments

The simulations presented in this paper have been carried out with the University of L'Aquila Climate-Chemistry Model (ULAQ-CCM), a CCM robustly tested and used before in evaluation of the radiative, chemical and dynamical effects of stratospheric and tropospheric aerosols (Pitari et al., 2002; Eyring et al., 2006; Morgenstern et al., 2010). It has also been used for various sulfate geoengineering simulations (Pitari et al., 2014; Visioni et al., 2018a, b) and, as part of the Climate-Chemistry Model Intercomparison project (Morgenstern et al., 2018), where it has been extensively validated with other CCMs. The high vertical resolution (127 levels) allows for a proper representation of large-scale transport of gas and aerosol species in the troposphere (Orbe et al., 2018) and in the stratosphere (Visioni et al., 2017b; Eichinger et al., 2019), and the detailed chemistry, including heterogeneous chemical reactions on sulfuric acid aerosols, polar stratospheric cloud particles, upper tropospheric ice and liquid water cloud particles allows for a full assessment of the effects of the increased sulfate burden on the atmospheric composition. ULAQ-CCM simulated COS also compares reasonably well with available measurements of seasonal COS concentrations (see Fig. S1) from Kuai et al. (2015), with an average annual error of 6.5%, albeit with peaks in some areas and months of up to 30%.

In addition to a reference historical model experiment (1960-2015), we performed four sets of simulations: a baseline unperturbed (BG) case and three geoengineering experiments (SG-COS-SRF, SG-COS-TRP and SG-SO2), all run between the years 2021-2055, with analyses focusing on the 2046-2055 decade; all experiments take place under the Representative Concentration Pathway 6.0 (RCP; Meinshausen et al., 2011) emissions.

The first geoengineering experiment, SG-COS-SRF, tries to produce a significant stratospheric aerosol burden by enhancing current anthropogenic emission of COS (0.12 Tg-S/yr, see table S1) by 40 Tg-S/yr. These emissions are located at the ground, in the main regions of anthropogenic COS surface emissions (see Fig. 1). The second experiment, SG-COS-TTL, tries to replicate the same stratospheric aerosol burden as SG-COS-SRF by injecting 6 Tg-S/yr of COS directly below the tropopause, at 16 km of altitude and at the equator. In the following text, whenever we are referring to results pertaining to both COS experiments, we will use the term SG-COS. Finally, the experiment SG-SO2, similarly to previous experiments discussed in

the literature (Kravitz et al. (2011) in the G4 experiment), consists of the injection of 4 Tg-S/yr in the form of $SO_2$ at the equator, between 18 and 25 km of altitude.

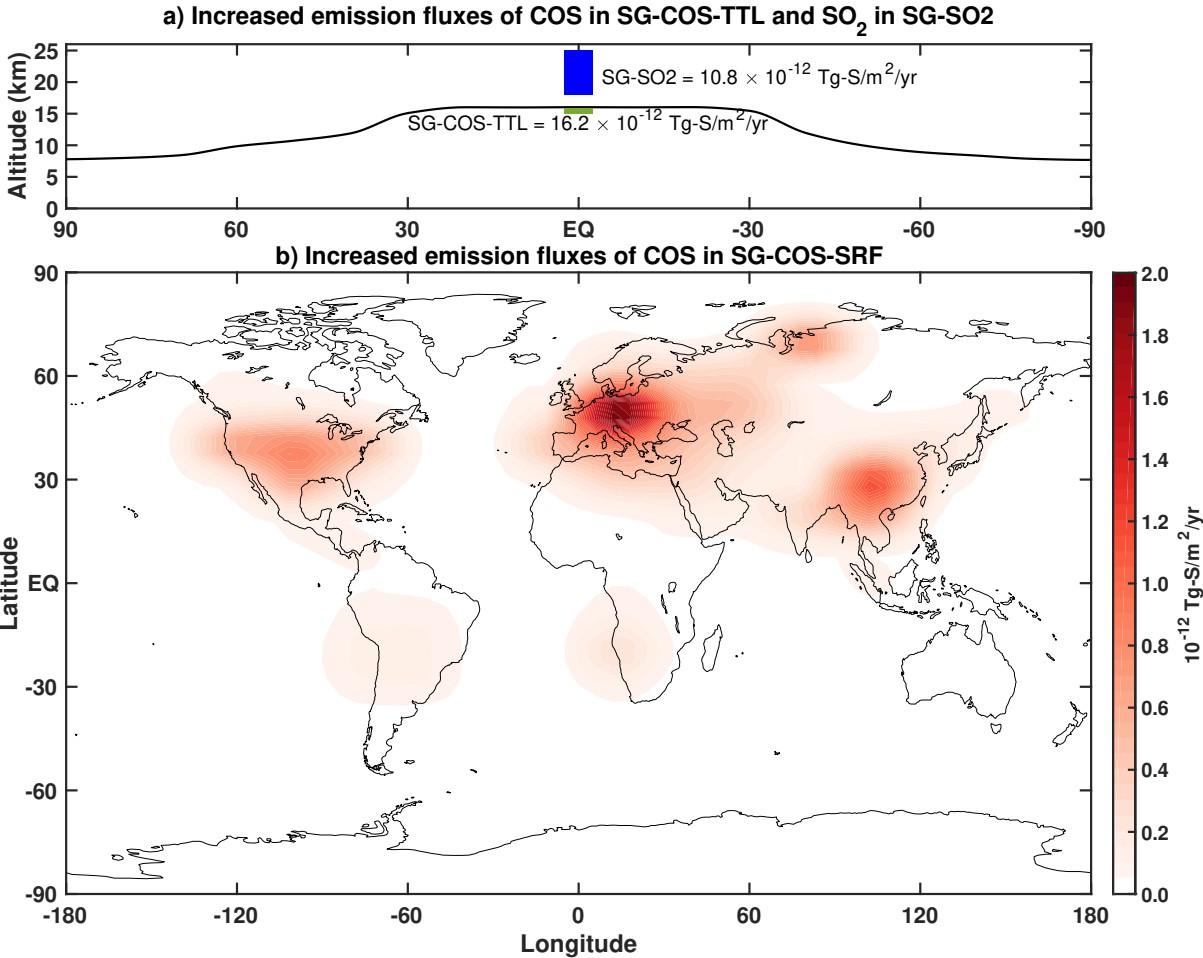

**Figure 1.** a) Vertical and latitudinal distribution of COS emissions per year and unit of surface area ($10^{-12}$ Tg-S/m$^2$/yr) in the SG-COS-TTL experiment (green box) and $SO_2$ emission fluxes in the same unit in SG-SO2 (blue box). The quantities are distributed in a single vertical level for SG-COS-TTL, and in 12 vertical levels for SG-SO2. b) Geographical distribution of COS emission fluxes per year and unit of surface area ($10^{-12}$ Tg-S/m$^2$/yr) in the SG-COS-SRF experiment. The annual upward flux is averaged over the period 2046-2055.

For the geoengineering experiments, ULAQ-CCM is driven by time-dependent sea surface temperatures (SSTs) from the Community Climate System Model – Community Atmosphere Model version 4 (CCSM-CAM4; Neale et al., 2013), an atmosphere-ocean coupled model that ran similar geoengineering experiments as in SG-SO2 (described by Tilmes et al., 2015): this allows for the inclusion of the cooling produced by geoengineering on the surface to the assessment of the dynamical and chemical effect as simulated by ULAQ-CCM. To include the important radiative effects produced by other atmospheric components (mainly, geoengineering-driven changes in greenhouse gases concentration and in ice clouds; Visioni et al., 2017b, 2018a)

the radiative module of ULAQ-CCM calculates at each time-step the surface temperature perturbation produced by the radiative flux changes induced by these components and includes them in the CCSM-CAM4 SSTs. This approach has been further explained and validated by Visioni et al. (2018a). While the prescribed SST setup has been shown to correctly capture dynamical changes produced by SRM (Visioni et al., 2017b), it clearly does not capture potential feedbacks that may be relevant for surface climate, such as those produced by the different latitudinal distribution of the aerosol optical depth that we will show later on. These differences may also in turn feed back onto changes in COS lifetime through precipitation changes (Whelan et al., 2016) which we can't consider here. We will therefore limit ourselves to analyzing changes in atmospheric composition and dynamics, and how those contribute to the overall radiative forcing from the aerosols. Future experiments with a more comprehensive Earth-system model will be necessary to determine the full extent of the climatic response.

## 3 Results

### 3.1 Sulfate burden

COS is the most abundant sulfur-containing species in the atmosphere under quiescent conditions (i.e. not considering explosive volcanic eruptions). It is efficiently lost at the surface via dry deposition on soils and vegetation: taking this sink into account, the net global lifetime (atmospheric chemistry plus surface deposition) is approximately 4 years, depending on the assumed magnitude of the soil and vegetation sink (Sandoval-Soto et al., 2005; Van Diest and Kesselmeier, 2008). In the troposphere the COS chemical reactivity (mostly with the hydroxyl radical) is rather slow: COS is thus well mixed and is easily transported in the stratosphere through the tropical tropopause layer (TTL). In the mid-stratosphere COS becomes efficiently photolyzed by solar UV radiation, becoming an important source for stratospheric $SO_2$ and finally for sulphuric acid aerosols.

When increasing the surface emission fluxes in SG-COS-SRF, it takes $\sim 15$ years before the concentration reaches a new equilibrium, from 0.5 to 35.5 ppbv (Fig. 2a), whereas in SG-COS-TTL the equilibrium value is 4.8 ppbv. In the same timespan, the global AOD increases reaching a value of 0.08 by 2035 in SG-COS-SRF and by 2030 in SG-COS-TTL, similar to the global value that is reached by the direct injection of $SO_2$ in the equatorial stratosphere in SG-SO2; in that case, however, the steady-state value is reached in only 1-2 years. In the GeoMIP G6sulfur experiment (Visioni et al. (2021b)), the average global surface cooling reported by 6 Earth system models for a similar stratospheric OD was 0.46 K. At the end of 2055, the increased COS and $SO_2$ injections are stopped. Average tropospheric COS concentrations follow an exponential decay guided by the atmospheric lifetime (3.8 years, due to chemistry but mainly soil deposition), reaching a value of 1.3 ppbv after 20 years in SG-COS-SRF (during 2075), whereas a similar value only takes 10 years to be reached in SG-COS-TTL. This means an increase of 0.8 ppbv with respect to background condition, that would produce a direct RF that is negligible compared to other well mixed greenhouse gases. The exponential decay of the stratospheric AOD in both SG-COS experiments is regulated by the stratospheric lifetime of COS (Fig. 2b), which is $\sim$10 years and it's mainly due to reaction with OH and photolysis, from which stratospheric $SO_2$ and finally sulphuric acid aerosols are formed. This is also combined with the depletion of the source of COS from the troposphere (Fig. 2a). Therefore, the e-folding time for stratospheric AOD is longer with respect to the one

resulting from SG-SO2 (Fig. 2c). In 2075, the global stratospheric AOD reaches a value of 0.01 in the SG-COS experiments

with respect to 0.003 in the background case.

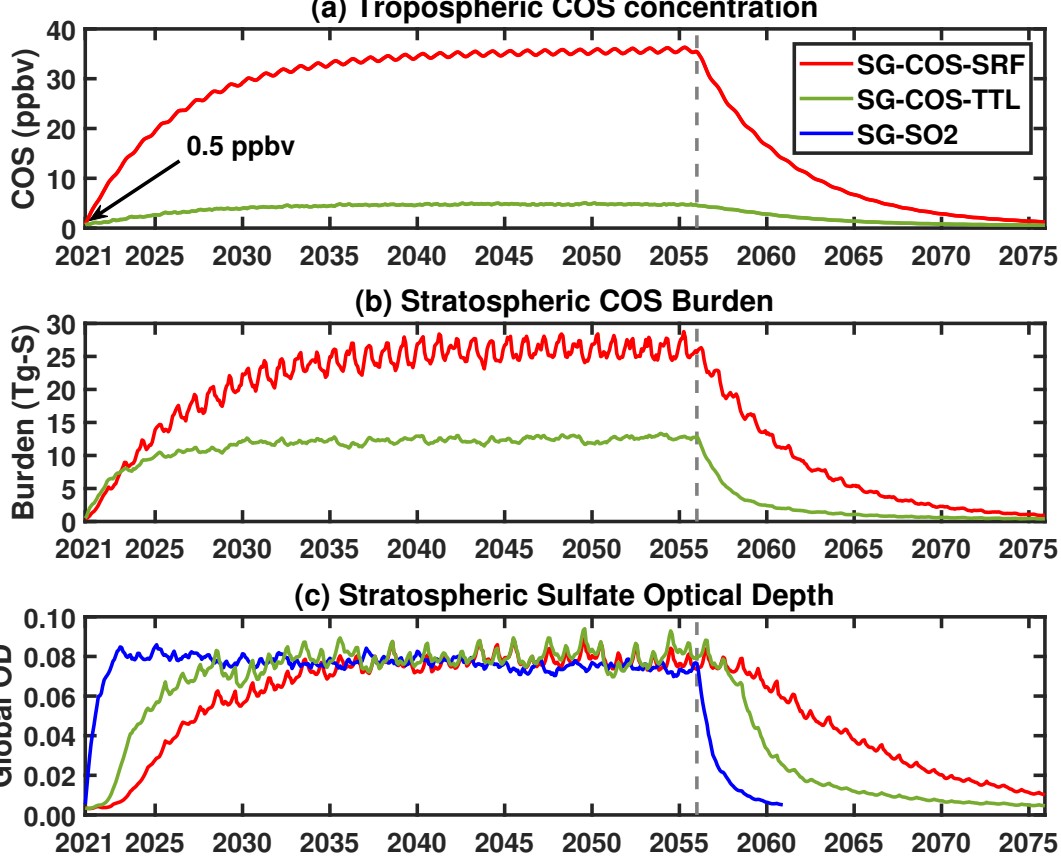

**Figure 2.** a) Monthly values of globally averaged tropospheric COS volume mixing ratio (ppbv) in both SG-COS experiments. The background value of 0.5 ppbv at the beginning of the simulation is highlighted. b) Monthly values of global stratospheric COS burden (in Tg-S) in both SG-COS experiments. c) Globally averaged stratospheric sulfate optical depth monthly values in SG-COS-SRF (red), SG-COS-TTL (green) and SG-SO2 (blue). The grey line in all panels indicates the time when emissions of COS and $SO_2$ are stopped, at the end of 2055.

## 3.2 Sulfate aerosol properties

| | BG | SG-COS-SRF | SG-COS-TTL | SG-SO2 | SG-COS-SRF - BG | SG-SO2 - BG |
|---|---|---|---|---|---|---|
| **Total Sulfate OD** | 0.054±0.003 | 0.134±0.005 | 0.134±0.004 | 0.128±0.004 | 0.080±0.005 | 0.074±0.005 |
| **Tropospheric Sulfate OD** | 0.051±0.003 | 0.056±0.003 | 0.054±0.003 | 0.054±0.003 | 0.005±0.0054 | 0.003±0.004 |
| **Stratospheric Sulfate OD** | 0.003±0.001 | 0.078±0.002 | 0.080±0.004 | 0.074±0.001 | 0.075±0.002 | 0.071±0.001 |
| **Sulfate effective radius ($\mu$m)** | 0.18±0.01 | 0.46±0.01 | 0.47±0.01 | 0.59±0.01 | | |
| **Ice OD** | 0.589±0.006 | 0.573±0.007 | 0.569±0.008 | 0.566±0.005 | -0.016±0.008 | -0.023±0.009 |
| **Ice effective radius ($\mu$m)** | 35±1 | 33±1 | 33±1 | 32±1 | | |

**Table 1.** Summary of calculated sulfate aerosol and cirrus ice globally-annually averaged quantities relevant for RF calculations (i.e., optical depth at $\lambda$=0.55 $\mu$m and effective radius). Last two columns show the calculated SG changes with respect to the BG case [years 2046-2055].

In both COS experiments, COS emissions are adjusted so as to have the same global aerosol optical depth (AOD) $\approx$ 0.08 (see table 1). This is done in order to more easily compare the latitudinal distribution of the aerosols, and to better quantify the differences in the radiative forcing from both direct and indirect (ozone, methane and water vapor) changes in atmospheric composition.

There is a large difference in the latitudinal distribution of stratospheric sulfate optical depth, as shown in figure 3 (a). Both SG-COS experiments produce an AOD more uniformly distributed over all latitudes with respect to the SG-SO2 case, where the increase of optical depth is most prominent in the tropics; this is due to the efficient tropospheric mixing of COS before it reaches the stratosphere even when, as in SG-COS-TTL, the injection happens close to the tropopause.

The differences in the latitudinal distribution of AOD are also observable in the differences in the particle sizes and in the surface area density (SAD). Figure 3 (b) shows that the stratospheric effective radius is smaller in the SG-COS experiments and uniform for all latitudes, with a global value of 0.46 $\mu$m. In SG-SO2, the effective radius is higher in the tropics (0.59 $\mu$m); AOD is also larger in the tropics in that case, due to a larger concentration of particles there, even if larger particles are less effective at scattering incoming solar radiation (English et al., 2012).

Figure 4 shows a comparison of the effective radius (a) and SAD (b) between the BG, SG-COS-SRF, SG-COS-TTL and SG-SO2 cases, separating the tropics, mid-latitudes and polar regions. As SO$_2$ is injected at the equator, all oxidation and nucleation happens in the tropics in SG-SO2. This is reflected in the vertical distribution which has a maximum in the lowermost stratosphere. On the other hand in SG-COS, the effective radius increase is reached at higher altitudes, between 18-30 km, which is consistent with COS reaching higher altitudes through deep tropical convection before it is photochemically destroyed (Barkley et al., 2008). The same explanation is valid for the tropical SAD in panel (b).

As the size of the particles is determined by nucleation in the tropical region, where SO$_2$ oxidation occurs, mid-latitude and polar behaviour of the aerosols depends on the poleward transport by the Brewer-Dobson Circulation (BDC).

In SG-SO2, aerosols grow rapidly in the tropical region due to the high concentration of SO$_2$, and their larger size affects sedimentation rates, thus decreasing their lifetime. Consequently, the amount of aerosols transported to higher latitudes is

lower; in SG-COS, smaller particles with a higher lifetime are either easily transported towards the poles or directly formed there. Smaller particles at a higher concentration, and larger particles at a lower concentration may then result in a SAD which looks similar at mid-latitudes and polar region, but for different reasons.

The vertical distribution of particles and their optical properties are shown in Figure 5 (see Fig. S2 for COS, $SO_2$ and $SO_4$ concentration changes; only values for one of the SG-COS experiments is shown here, as they are indistinguishable).

The vertical distribution of the SAD is fundamental to understand the role of the heterogeneous reaction and their effect on stratospheric ozone. The baseline case in panels (a) and (b) are a reference for understanding their changes in the SG-COS experiments (panels (c) and (d)). The particles transported via the BDC to the poles are large enough to efficiently scatter solar radiation so that the SAD and extinction changes show a similar behaviour, with a global increase of stratospheric values with maxima at higher latitudes between 15-25 km.

Panels (f) and (d) show the extinction and SAD changes between the three SG experiments, to underline that in SG-SO2 extinction of the radiation is confined in the tropical stratosphere between 15-25 km, meaning a negative change in SG-COS. As discussed before, the formation of larger particles in SG-SO2 in the tropical region reduces the amount of aerosol transported to the poles compared to the SG-COS cases, where a larger number of smaller particles produces a positive change in SAD and, consequently, in extinction.

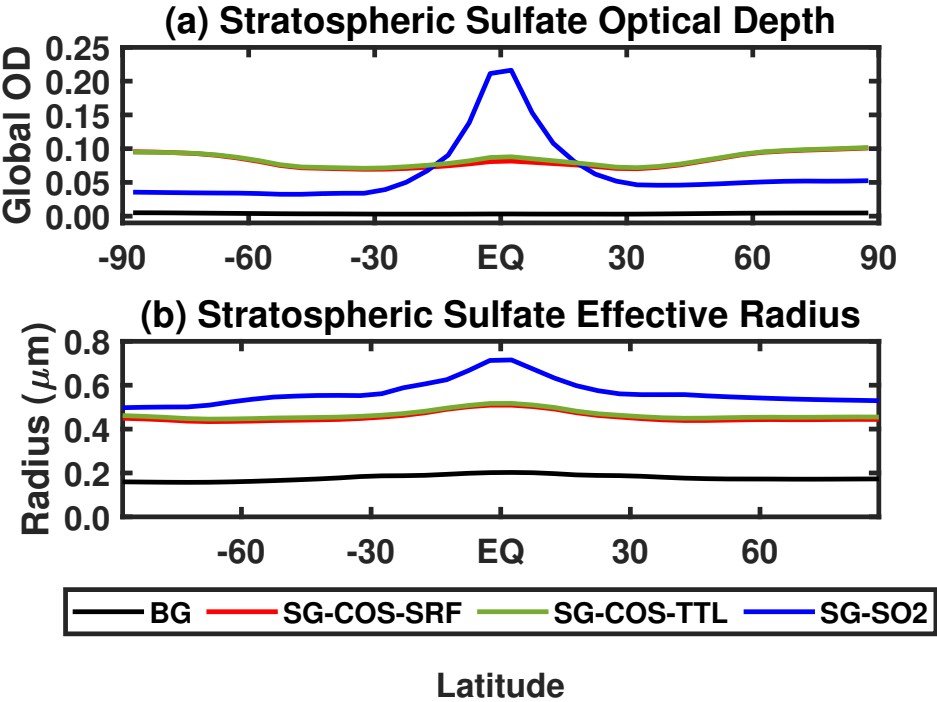

**Figure 3.** a) Latitudinal distribution of zonal mean values of stratospheric sulfate optical depth for the BG (black), SG-COS-SRF (red), SG-COS-TTL (green) and SG-SO2 (blue) cases. b) Stratospheric effective radius (in $\mu$m, from tropopause to 6hPa). All quantities are annually averaged over the years 2046-2055.

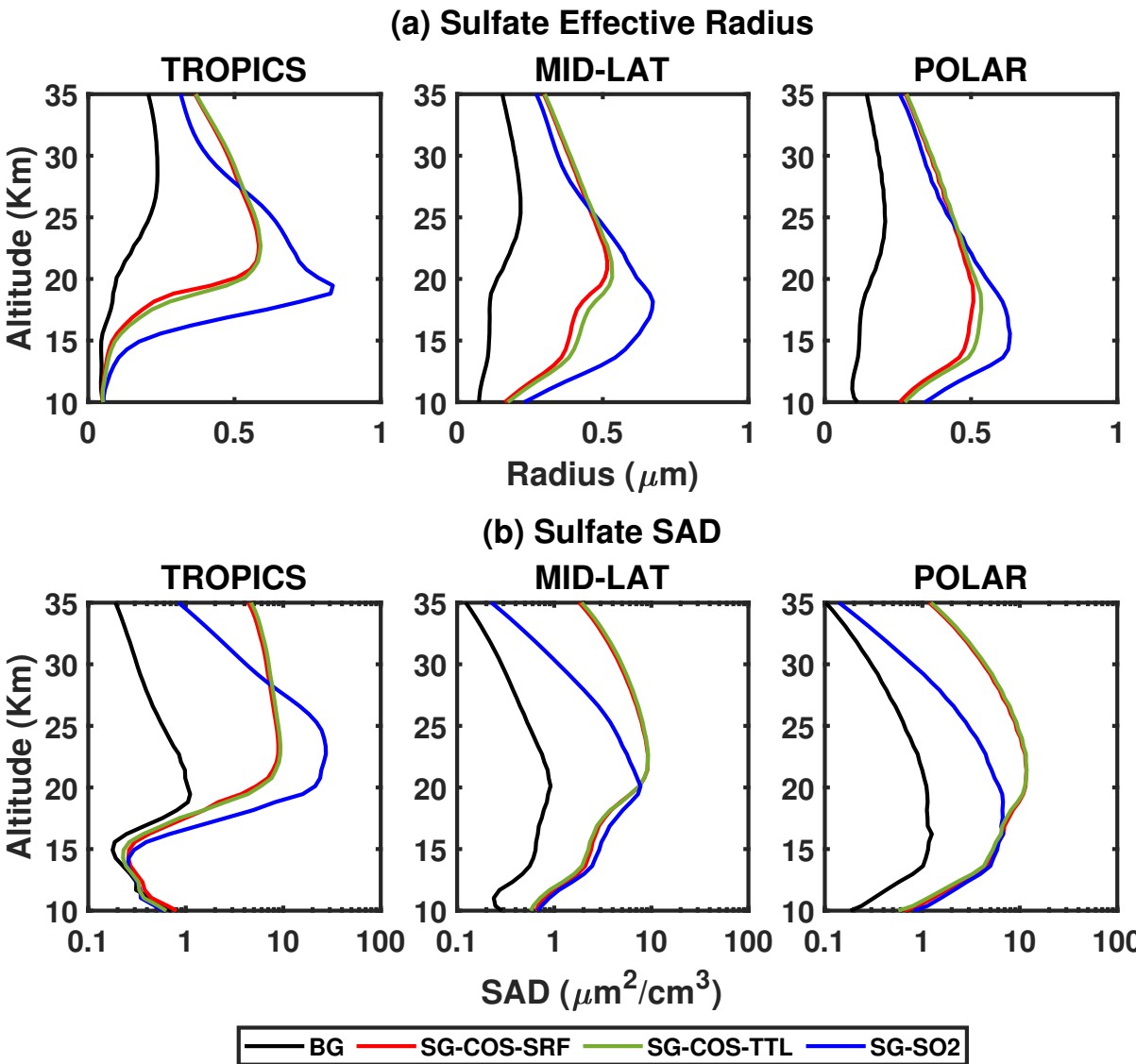

**Figure 4.** Vertical profiles of sulfate effective radius (in $\mu$m, a) and surface area density (in $\mu$m$^2$/cm$^3$, b) at different latitudinal bands (20N-20S for the Tropics, 30-50 at both N and S for the Mid-lat and 60-90 at both N and S for the Polar plots). All quantities are annually averaged over the years 2046-2055.

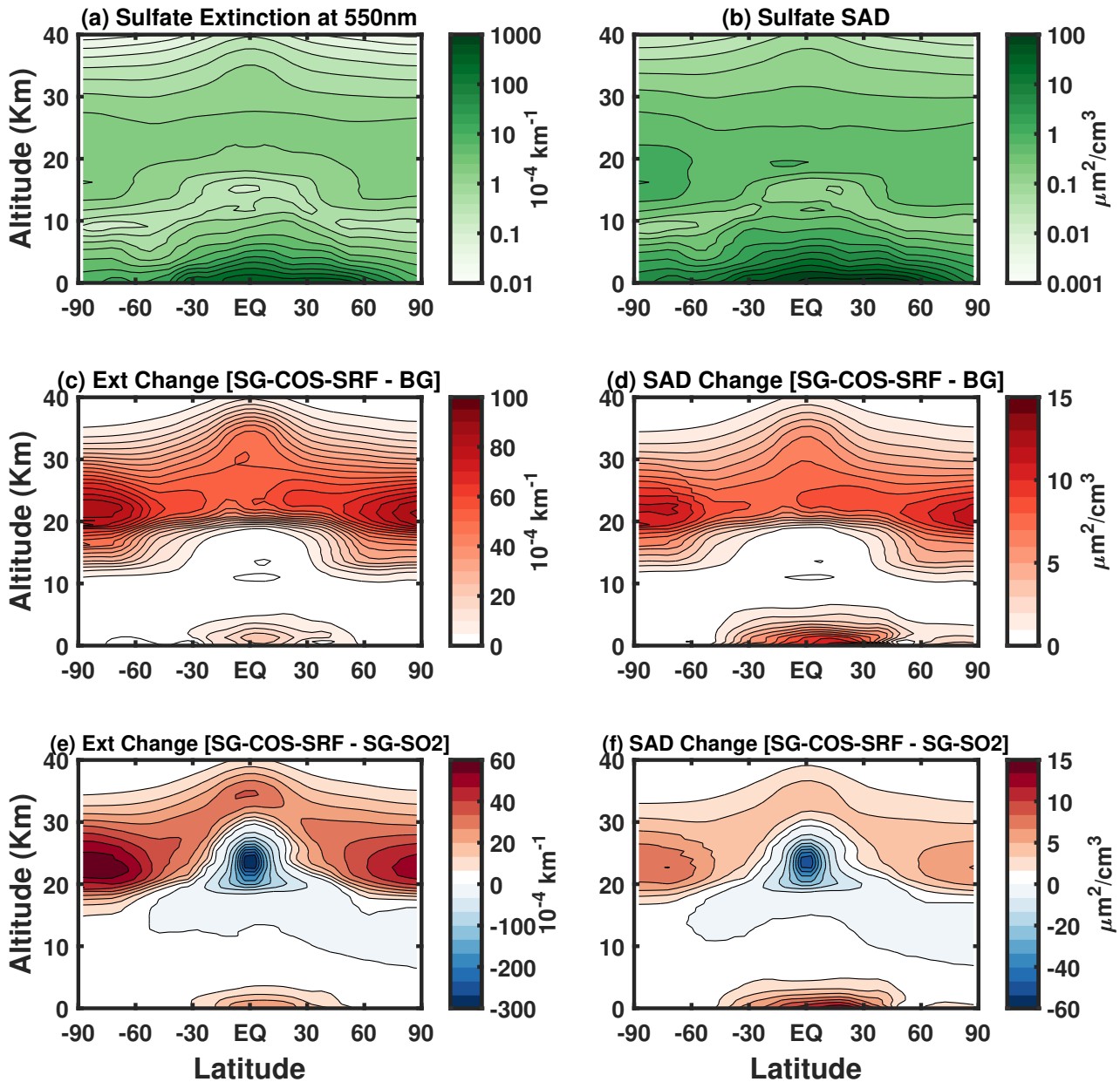

**Figure 5.** Zonal mean values of sulfate extinction (in $10^{-4}$ km$^{-1}$) and SAD (in $\mu$m$^2$/cm$^3$) in BG (panels a and b, respectively) and their change in the case of the SG-COS-SRF experiment (panels c and d). Panels e) and f) show extinction and SAD changes between SG-COS-SRF and SG-SO2. All quantities are annually averaged over the years 2046-2055.

 **3.3 Deposition**

| Experiment | MSA | $SO_2$ | $SO_4$ | COS | $CS_2$ | H2S | Total |
|---|---|---|---|---|---|---|---|
| **BG** | 1.0±0.1 | 35.2±1.4 | 9.4±0.4 | 0.39±0.01 | 0.47±0.03 | 1.5±0.1 | 48.0±1.8 |
| **SG-COS-SRF** | 1.0±0.1 | 36.4±1.5 | 9.9±0.4 | 31.6±0.1 | 0.47±0.03 | 1.5±0.1 | 80.9±1.7 |
| **SG-COS-TTL** | 1.0±0.1 | 35.8±1.5 | 9.7±0.4 | 3.5±0.1 | 0.47±0.03 | 1.5±0.1 | 52.0±1.7 |
| **SG-SO2** | 1.0±0.1 | 35.6±1.5 | 9.5±0.4 | 0.39±0.01 | 0.47±0.03 | 1.5±0.1 | 48.5±1.8 |

**Table 2.** Globally-annually averaged dry deposition rates of sulfur species (Tg-S/yr) [years 2046-2055].

| Experiment | MSA | $SO_2$ | $SO_4$ | Total | Net [sources-sinks] |
|---|---|---|---|---|---|
| **BG** | 1.5±0.1 | 3.0±0.1 | 43.2±1.5 | 47.7±1.6 | +0.3±0.1 |
| **SG-COS-SRF** | 1.5±0.1 | 3.4±0.1 | 49.4±1.5 | 54.3±1.6 | +0.8±0.1 |
| **SG-COS-TTL** | 1.5±0.1 | 3.2±0.1 | 45.2±1.5 | 49.9±1.6 | +0.1±0.1 |
| **SG-SO2** | 1.5±0.1 | 3.2±0.1 | 46.5±1.5 | 51.2±1.6 | +0.3±0.1 |

**Table 3.** Globally-annually averaged wet deposition rates of sulfur species (Tg-S/yr) [years 2046-2055]. The last column shows the net balance of total sulfur sources and sinks (Tg-S/yr).

The enhancement of the stratospheric sulfate burden would produce an increase in sulfur deposition, in dry form through acid gas deposition and in wet form through rain, fog and aerosol particles.

Acid deposition may damage human health when high concentrations of particles with a diameter below certain thresholds (PM2.5 and PM10) are inhaled. The acidification of soils and water may damage plants, microorganisms and aquatic animals, but the impact on the ecosystem depends on the rate at which acidifying compounds are deposited from the atmosphere, compared with the rate at which acid neutralizing capacity is generated within the ecosystem (Driscoll et al., 2001).

Here we analyse how dry and wet deposition of sulfur species are distributed globally as a result of the two SG interventions. Table 2 and 3 summarize wet and dry deposition rates for the SG-COS, SG-SO2 and BG experiments, and they include the contribution of each species to the total deposition. In particular, in both SG-COS experiments the increase in COS fluxes produces both an increase in sulfuric deposition, after its photolysis and oxidation to sulfuric acid, and in dry deposition of COS itself, as it is removed to the ground through uptake by vegetation and soils (Kettle et al., 2002).

The global distribution of COS deposition for the baseline case is shown in figure 6(a) while the increase in deposition from the SG-COS-SRF experiment is shown in figure 6(b). For the SG-COS-TTL case, the spatial distribution is identical to

SG-COS-SRF, but its magnitude is 10 times lower than in SG-COS-SRF. COS uptake by plants is concentrated mainly in the
tropical rainforests of South America, Africa, and southeast Asia and boreal coniferous forests across North America, northern
Europe, and northern Asia. Uptake by soils occurs mainly in arid and semiarid regions, such as savanna regions in the northern
and southern Africa and in the southwestern regions of North America, in the pampas of Argentina, in Australia, and in the
steppes of central Asia (Kettle et al., 2002). Dry deposition of COS doesn't contribute to acid deposition and, currently, there is
no information available on how different soils or ecosystems would be affected by higher local COS concentrations; therefore,
we assumed that their uptake efficiency does not change. The robustness of this assumption will need to be studied.

The global distribution of $SO_x$ deposition is also shown in figure 6. Panels (c) and (d) show dry and wet deposition, respec-
tively, for the background case. Dry deposition maxima are localized in urban areas close to the source where the emitted sulfur
dioxide is immediately oxidized, while wet deposition distribution depends both on sulfate concentration and precipitation.

Panels (e) and (f) show the total $SO_x$ deposition change in SG-COS-SRF with respect to the baseline case, in absolute terms
and as a percentage of the baseline case, and most of its increase is due to wet deposition (see tables 2 and 3, and see Tables S1-
4 for a breakdown of global sources and sinks of sulfur species). In both figures, the distribution of deposition is more uniform
over the globe with respect to the tropical injection of $SO_2$, except for the polar regions, because of the reduced precipitation
rates. Consequently, figure 6 (f) shows a large increase in percent deposition in the polar region (17% in the Arctic, 8% in
Antarctic; these values are reduced to 1.7% and 0.8 % in SG-COS-TTL, see Fig. S4) because of very low values in the baseline
case. On the other hand, deposition change is close to zero in polluted regions.

Globally, the annual differences in deposition fluxes for all species compared to the background case amount to $8.3 \pm 0.2$
Tg-S/yr for SG-COS-SRF, $3.1 \pm 0.2$ Tg-S/yr and $3.9 \pm 0.2$ Tg-S/yr for SG-SO2, which equates to an increase of $8.9 \pm 0.3$ %,
$3.3 \pm 0.3$ % and $4.2 \pm 0.3$ %, respectively.

## 4  Indirect effects

| | BG | SG-COS-SRF | SG-COS-TTL | SG-SO2 | SG-COS-SRF - BG | SG-COS-TTL - BG | SG-SO2 - BG |
|---|---|---|---|---|---|---|---|
| **COS [troposphere] (ppbv)** | 0.47±0.1 | 35.5±0.2 | 4.8±0.1 | 0.47±0.1 | 35.0±0.2 | 4.3±0.2 | 0.00±0.1 |
| **CH$_4$ lifetime (yr)** | 8.72±0.13 | 9.83±0.18 | 9.85±0.17 | 9.78±0.20 | 1.11±0.13 [(+12.7±1.4) %] | 1.13±0.13 [(+13.0±1.4) %] | 1.06±0.17 [(+12.2±2.0) %] |
| **H$_2$O [stratosphere] (ppmv)** | 6.08±0.08 | 5.99±0.16 | 5.95±0.15 | 6.13±0.13 | -0.09±0.14 | -0.13±0.15 | 0.05±0.12 |
| **O$_3$ column (DU)** | 289.3±1.8 | 294.2±1.5 | 294.8±1.6 | 290.7±1.6 | 4.9±2.3 | 5.5±2.4 | 1.4±1.7 |

**Table 4.** Summary of calculated globally-annually averaged quantities of greenhouse gases directly and indirectly perturbed by SG and
relevant for RF calculations (i.e., COS mean tropospheric mixing ratio, CH$_4$ atmospheric lifetime, H$_2$O mean stratospheric mixing ratio, O$_3$
column). Last two columns show the calculated SG changes with respect to the BG case [years 2046-2055].

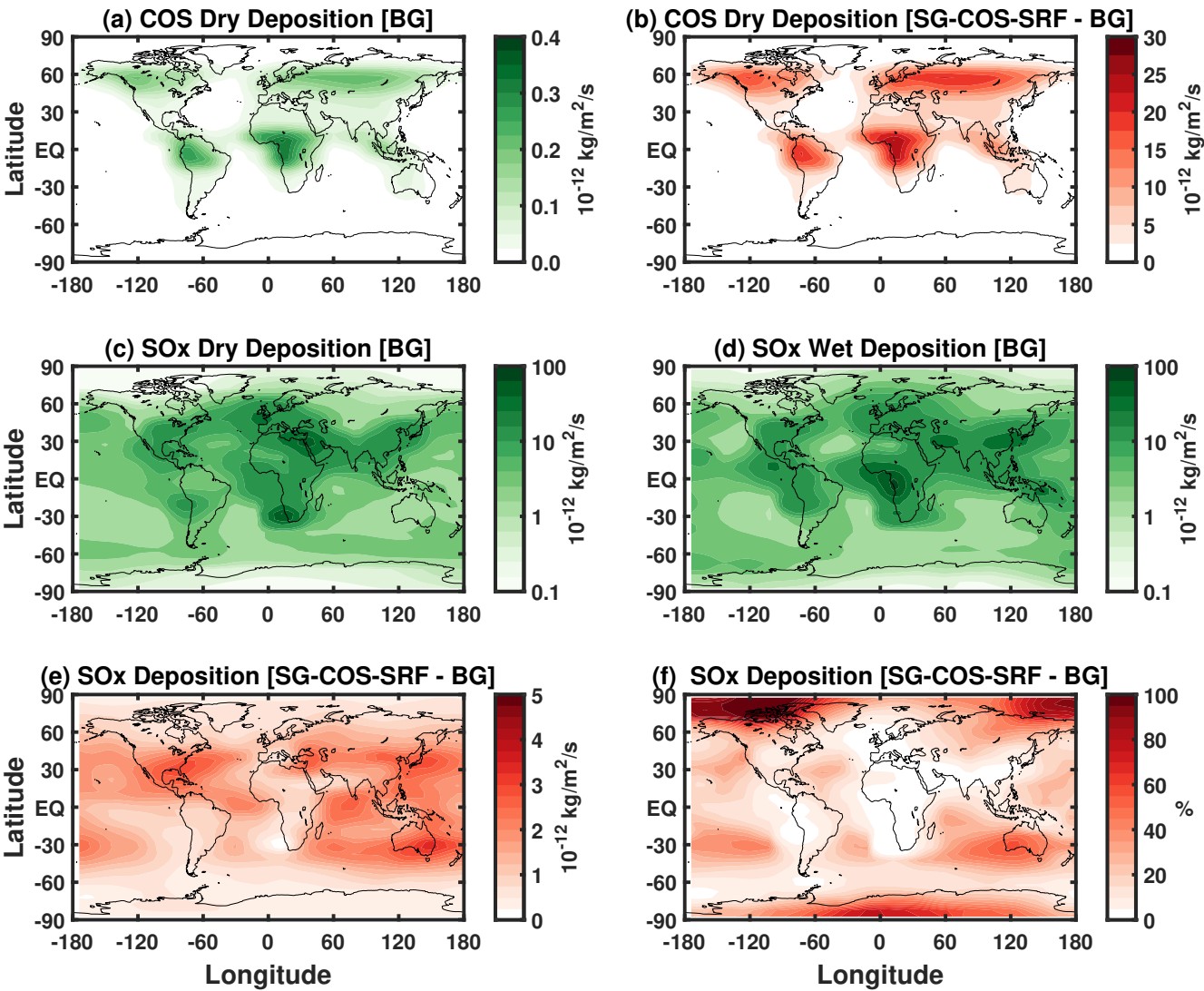

**Figure 6.** a) Surface dry deposition fluxes ($10^{-12}$kg/m²/s) of COS in the background case. b) Change in COS dry deposition fluxes in SG-COS-SRF compared to a). c) SO$_x$ dry deposition fluxes ($10^{-12}$kg/m²/s) in the background case. d) SO$_x$ dry deposition fluxes ($10^{-12}$kg/m²/s) in the background case. e) Change in SO$_x$ total deposition fluxes in SG-COS-SRF compared to the background. f) as e) but in % of the background values.

| | BG (W/m$^2$) | SG-COS-SRF (W/m$^2$) | SG-COS-TTL (W/m$^2$) | SG-SO2 (W/m$^2$) | SG-COS-SRF - BG (%) | SG-COS-TTL - BG (%) | SG-SO2 - BG (%) |
|---|---|---|---|---|---|---|---|
| UVB | 0.206±0.002 | 0.197±0.001 | 0.196±0.001 | 0.201±0.001 | -4.4±0.6 | -5.8±0.6 | -2.4±0.9 |
| UVA | 11.35±0.01 | 11.13±0.01 | 11.12±0.01 | 11.17±0.01 | -1.9±0.1 | -2.0±0.1 | -1.6±0.1 |

**Table 5.** Summary of calculated globally-annually averaged quantities of UVB and UVA at surface. Last two columns show the calculated SG percentage changes with respect to the BG case [years 2046-2055].

The simulated enhancement in the stratospheric aerosol layer would produce two main effects: an increased scattering of solar radiation, that in turn would reduce surface temperatures, and the local absorption of more near-infrared solar and terrestrial radiation, that would warm the stratospheric layer where the aerosols reside (as observed for volcanic eruptions, see Lacis et al., 1992; Labitzke and McCormick, 1992). Furthermore, the increase in the surface area density of the aerosols would affect the heterogeneous chemistry of $ClO_x$ and $NO_x$ with implications for ozone concentration and UV radiation at the surface (Tilmes et al., 2009, 2018b, 2021).

For $SO_2$, it has been shown that the combination of surface cooling, perturbation of stratospheric temperatures and changes in tropospheric ozone and in UV at the surface also affects methane lifetime (Visioni et al., 2017a). In this section we analyse the differences in these changes also for the SG-COS-SRF experiment.

Figure 7 shows the ozone changes in SG-COS-SRF and SG-SO2 with respect to the BG case. As expected from the similar value and distribution of the SAD, in SG-COS-TTL ozone changes are equivalent to SG-COS-SRF (and are therefore not shown). Panels (a) and (b) show the monthly total ozone column changes as a function of latitude. Close to the equator there is a small reduction in the overall column, mostly due to a reduction in tropospheric ozone, as visible in panels (c) and (d), as a direct consequence of the surface cooling (Nowack et al., 2016). On the other hand, at higher latitudes an overall increase in the total column is observable due to an increase in stratospheric ozone. This is particularly evident closer to the poles.

During springtime months, there is some Antarctic ozone depletion, while in the Arctic a recovery of ozone is observable. In the Antarctic spring, the polar vortex is strengthened by the stratospheric heating in the tropics that affects the equator-to-pole thermal wind balance (Visioni et al., 2020), resulting in greater confinement of cold air, that, in turn, enhances the ozone depletion by the polar stratospheric clouds (PSC). The tropical stratospheric heating is higher in SG-SO2 with respect to SG-COS as the aerosols are less confined (fig. S6). Consequently, the strengthening of polar vortex in SG-SO2 produces a higher ozone depletion. In the Arctic, on the other hand, PSC-related ozone loss is lower (Tilmes et al., 2018a), and the predominant effect is that from an acceleration of the BDC transporting ozone-rich air from lower latitudes.

Panels (c) and (d) show the annual mean of ozone mixing ratio percentage change as a function of altitude and latitude. In both SG experiments, negative changes below the tropopause are governed by the decrease in solar radiation which comes into play in the photo-dissociation reaction of $NO_x$ as an ozone precursor ($NO_2 + h\nu$ ($\lambda < 420$ nm) $\rightarrow NO + O(^3P)$). Sunlight reduction also affects the $O_3$ photolysis, decreasing the ozone loss. Positive changes are due to the balance of the previous reactions and the increase of methane (see table 4) as a source of ozone in its oxidation chain, and mainly due to the decrease

of the tropospheric water vapour in a clean air environment (low $NO_x$), such as the tropics (Nowack et al., 2016; Xia et al., 2017b).

Above the tropopause, there is a negative ozone change in the lower stratosphere in all SG experiments, except for the Arctic region where we observe a small increase in the Arctic lowermost stratosphere in all cases. The key drivers of stratospheric ozone change are the increase in heterogeneous reactions, as a result of the enhancement of stratospheric aerosols, and the perturbation of the dynamics governing ozone transport.

Negative ozone changes correspond to the region where the SAD reaches its maximum values (fig. 5 (d) and (f)): between 10-20 km in the polar regions for SG-COS and mainly between 15-25 km at tropics for SG-SO2. The increase of the SAD enhances heterogeneous chemistry and results in denitrification via hydrolysis of dinitrogen pentoxide ($N_2O_5 + H_2O \xrightarrow{M}$ $2HNO_3$). The loss of $NO_x$ decreases the rate of ozone depletion through its catalytic cycle. Whereas in the mid-stratosphere, where the cycles of chlorine ($ClO_x$) and bromine ($BrO_x$) are dominant, there is an increase in ozone loss since reduction of $NO_x$, that normally bounds chlorine ($ClONO_2$), allows ClO to destroy more ozone (Tilmes et al., 2018b; Grant et al., 1992).

At low latitudes, stratospheric ozone concentration is also driven by changes in tropical upwelling (Visioni et al., 2021a): the reduction in tropical upwelling of ozone-poor air coming from the lowermost stratosphere leads to higher ozone concentration at altitudes of about 20-22 km (Tilmes et al., 2018b).

Figure S7 (e) shows the change of tropical upwelling in relation to changes in the residual vertical velocity ($w^*$) with respect to the baseline case. Negative $w^*$ anomalies in SG-COS mean weaker tropical upwelling as consequence of tropospheric cooling. In SG-SO2, the highest concentration of absorbing aerosols leads to positive $w^*$ above 20 km due to the local warming but this doesn't affect the transport of ozone-poor air from the lower layers.

Above the discussed altitudes, there is a net ozone production in all SG experiments, with an higher increase of ozone mixing ratio in SG-COS experiment with respect to SG-SO2, especially in the extra-tropical region. Ozone depletion at these altitudes is mainly controlled by the catalytic cycle of $NO_x$, that is inhibited by the denitrification process due to heterogeneous reactions on aerosols.

Globally, the annually-averaged ozone column increases of ~5 and 1.5 DU for SG-COS and SG-SO2, respectively (table 4). Increasing stratospheric ozone affects UVB at the surface because it is absorbed by ozone during its photodissociation, while aerosol could affect UVA radiation by scattering processes: the projected changes are shown in figure 8 for both UVA and UVB for each season and for the annual mean. We estimated these changes using TUV (from https://www2.acom.ucar.edu/ modeling/tropospheric-ultraviolet-and-visible-tuv-radiation-model), using in input our model latitudinal and monthly values for the period (2046-2055) for aerosol optical depth, total ozone column, climatological cloud cover and surface albedo.

In all SG experiments, the negative changes of UVB radiation at the surface, except in the Antarctic region, are related to changes in stratospheric ozone, as well as the interannual variations that are larger at the poles, due to the seasonal variability, as discussed before. In the Antarctic Spring (SON) the ozone depletion is enhanced in SG-SO2 while in SG-COS-SRF it is limited to the month of October, with differences compared to BG of less than -5 DU. Therefore, the UVB change compared to BG for SON over Antarctica remains negative in SG-COS-SRF with a value of -2.7% versus a +5.8% increase in the SG-SO2 experiment. In DJF, on the other hand, a small increase of UVB is observable at mid to high latitudes in the Northern

Hemisphere. This is connected to an observable decrease of stratospheric ozone in the same locations, possibly due to a reduced advection of air from the tropics. UVA decreases everywhere in all SG experiments. In particular, the correlation between UVA change and particles scattering is evident if we compare this latitudinal distribution with the stratospheric AOD of figure 3(a). The globally averaged UVB and UVA changes at surface are summarized in table 5.

Methane is an indirect source of tropospheric ozone (West and Fiore, 2005), and it is also a greenhouse gas. Knowing its variation is fundamental to understand the final contribution to the radiative forcing that one would wish to achieve with this geoengineering method. From table 4, we find a global increase in methane lifetime of $\sim$ 13% in SG-COS and 12.2% in SG-SO2, which we can identify in the increase in methane itself. The reason for the increase in methane is to be found in the behaviour of the hydroxyl radical (OH), as the main sink of methane is the oxidation reaction with OH: decrease of OH means an increase of methane lifetime. As discussed by Visioni et al. (2017b), mechanisms that cause an increase in OH are as follows: (a) surface cooling lessens the amount of tropospheric water vapor and inhibits the temperature-dependent reaction of $NO + O_3$; (b) decrease of tropospheric UV, due to enhancement of ozone and scattering radiation, reduce $O(^1D)$ that takes part of the reaction $O(^1D) + H_2O \rightarrow 2\ OH$; (c) increase of SAD enhances heterogeneous chemistry reducing the amount of $NO_x$ ($NO + HO_2$, $NO + RO_2$); (d) increase of tropical lower stratosphere temperature (TTL) that regulates the stratosphere-troposphere exchange, which can be positive or negative depending on the net result of the superimposed species ($CH_4$, $NO_y$, $O_3$, $SO_4$) in the extratropical upper troposphere-lower stratosphere (UTLS).

The warming of the TTL is shown in figure S7 (d): in SG-SO2, larger particles confined in the tropical region produce a greater warming of the TTL with respect to smaller ones distributed all over the globe in SG-COS. The role of dimensions and distributions of aerosols in stratospheric warming is confirmed by the heating rates, as shown in figure S6.

## 5  Radiative forcing

| Total RF (W/m$^2$) | SW | LW | NET |
|---|---|---|---|
| SG-COS-SRF | -1.47±0.12 | +0.21±0.25 | -1.26±0.13 |
| SG-COS-TTL | -1.41±0.12 | -0.06±0.25 | -1.47±0.13 |
| SG-SO2 | -1.58±0.10 | -0.11±0.23 | -1.69±0.13 |

Table 6. Globally-annually averaged total RF of sulfate aerosols and greenhouse gases for the SG experiments with respect to BG (shortwave, longwave and net) (W/m$^2$) [years 2046-2055].

The ULAQ-CCM radiative transfer module calculates online the radiative forcing due to aerosols, greenhouse gases (GHGs), and low and high clouds. The effects of single components have been estimated offline for both shortwave (SW) and longwave (LW) with the same radiative transfer core, for sulfate aerosols, clouds, COS, $CH_4$, stratospheric $H_2O$, stratospheric and tropospheric $O_3$ in order to properly separate the contributions.

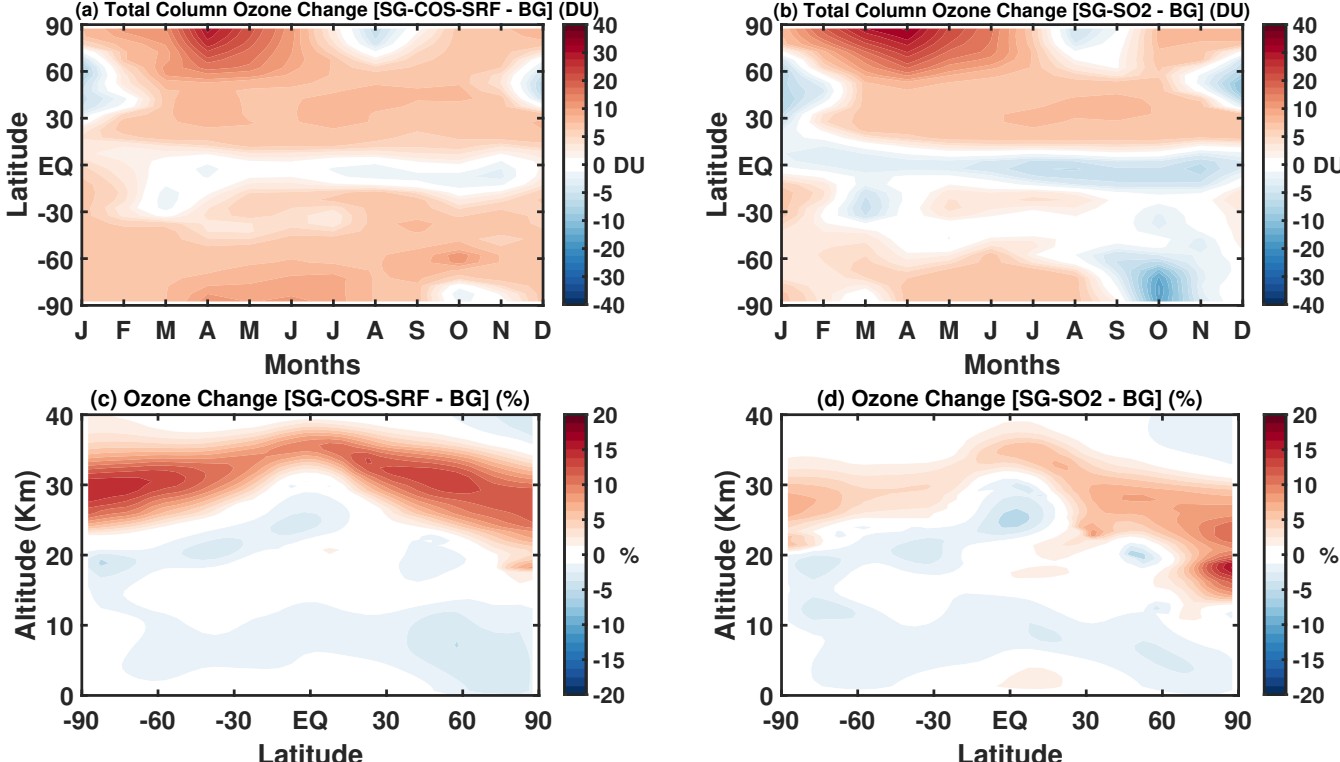

**Figure 7.** a,b) Monthly mean zonal values of SG ozone total column changes (DU) with respect to the BG case for SG-COS-SRF and SG-SO2 respectively. c, d) ozone mixing ratio percent changes with respect to the BG case. All quantities are annually averaged over the years 2046-2055.

Tables S8, S9 and S10 summarise the individual contributions of GHGs changes for SG-COS-SRF, SG-COS-TTL and SG-SO2, respectively. Similar increases of methane in all SG experiments produce the same positive LW RF; the TTL warming (which results in an increase in stratospheric water vapor), results in a small but positive contribution from $H_2O$ in SG-SO2. Contributions from both stratospheric and tropospheric $O_3$ changes have also been estimated, but are negligible.

In both SG-COS experiments, obviously, the increase of COS concentration, which is a GHG, must be taken into account. We estimated its contribution to the radiative forcing based on the definition of global warming potential (GWP) on a mass/mass basis as in Brühl et al. (2012) for a time horizons of 30 years (2021-2050). GWP can be approximated as follows by the expression of Roehl et al. (1995), assuming that the perturbation of the radiation balance of the Earth by greenhouse gases COS and $CO_2$ decays exponentially after a pulse emission for a time horizon $\Delta T$.

$$GWP_{\Delta_t} \simeq \frac{RF_{COS}}{RF_{CO_2}} \times \frac{\tau_{COS}}{\tau_{CO_2}} \times \frac{1 - e^{\frac{-\Delta_t}{\tau_{COS}}}}{1 - e^{\frac{-\Delta_t}{\tau_{CO_2}}}} \tag{1}$$

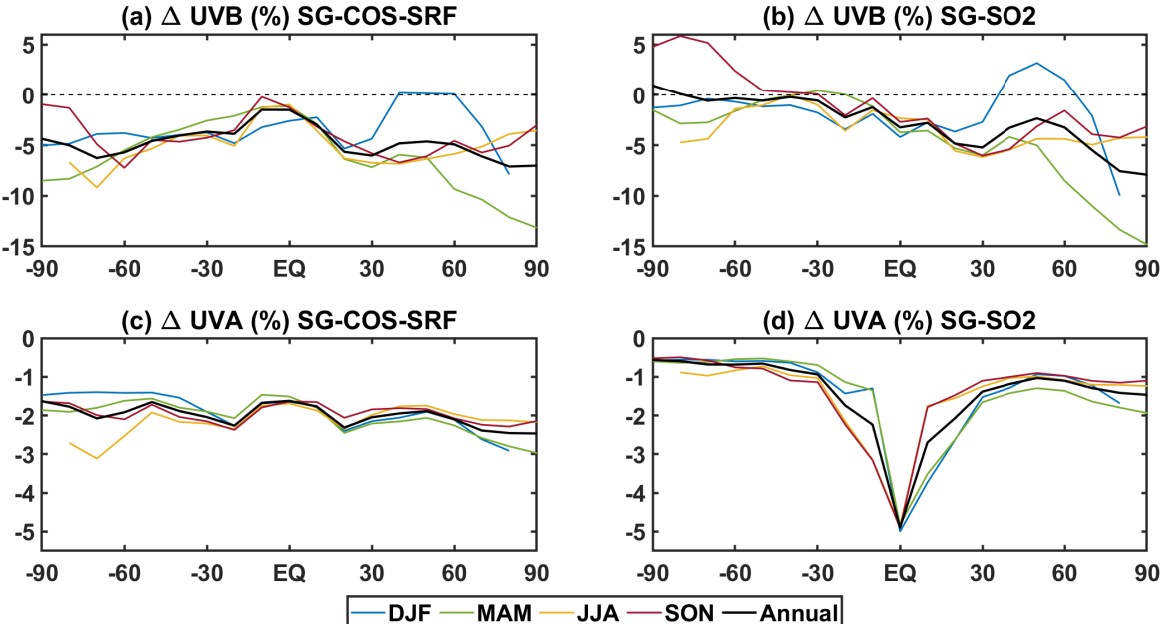

**Figure 8.** Zonal UVB and UVA surface changes per each season in percentage with respect to BG case in SG-COS-SRF (panels a and b, respectively) and SG-SO2 (panels c and d). All quantities are averaged over the years 2046-2055.

We assumed an overall lifetime of $\tau_{COS}$=3.8 yr and $\tau_{CO_2}$=75 yr, and the radiative forcing of 1 kg of COS relative to 1 kg of $CO_2$ added to the present atmosphere ($RF_{COS}/RF_{CO_2}$) is 724 (Brühl and Crutzen, 1988). This results in a GWP of 111. For our time period, the mass of COS and $CO_2$ added to the atmosphere ($\Delta m$) is $1.97 \times 10^{12}$ kg of COS (for SG-COS-SRF), 0.35 $\times 10^{12}$ kg of COS (for SG-COS-TTL) and $1.23 \times 10^{15}$ kg of $CO_2$. Therefore, the COS radiative forcing can be calculated as:

$$RF_{COS} = GWP_{\Delta t} \times RF_{CO_2} \times \frac{\Delta m_{COS}}{\Delta m_{CO_2}} \tag{2}$$

where $RF_{CO_2}$ in RCP6.0 is estimated to be 0.83 W/m$^2$ considering an increase of 68.5 ppm from a baseline of 409.2 ppm. Overall, this results in a radiative forcing from the COS increase of 0.17 W/m$^2$ in SG-COS-SRF and of 0.03 W/m$^2$ in SG-COS-TTL.

The main contributions of sulfate aerosols and clouds are summarised in tables S5, S6 and S7 for SG-COS-SRF, SG-COS-TTL and SG-SO2, respectively. The contribution of sulfate aerosols is the sum of the cooling effects given by the efficient scattering of solar radiation by particles of radius of around 0.5 $\mu$m and the absorption of LW by larger ones. Globally, the estimated values are similar for the Clear-Sky SW and LW forcing from the sulfate aerosols: in terms of the latitudinal distribution, however, SG-SO2 presents a peak in the tropics whereas the forcing from SG-COS is much more latitudinally even.

The reduction in optical depth from cirrus clouds (see table 1) produced by the aerosols (Kuebbeler et al., 2012; Visioni et al., 2018a) results in a net negative radiative forcing. This is given by the balance between the positive RF in the shortwave (SW) due to the reduction of reflected solar radiation and the negative RF in the longwave (LW) due to the decrease of trapped planetary radiation, which reduces the contribution to the greenhouse effect. In the SG-COS cases, at the equator the positive RF from the thinning locally balances the direct forcing from the aerosol (figures 9 and S8).

Table 6 summarises the total contribution of sulfate aerosols and greenhouse gases under All-Sky conditions.

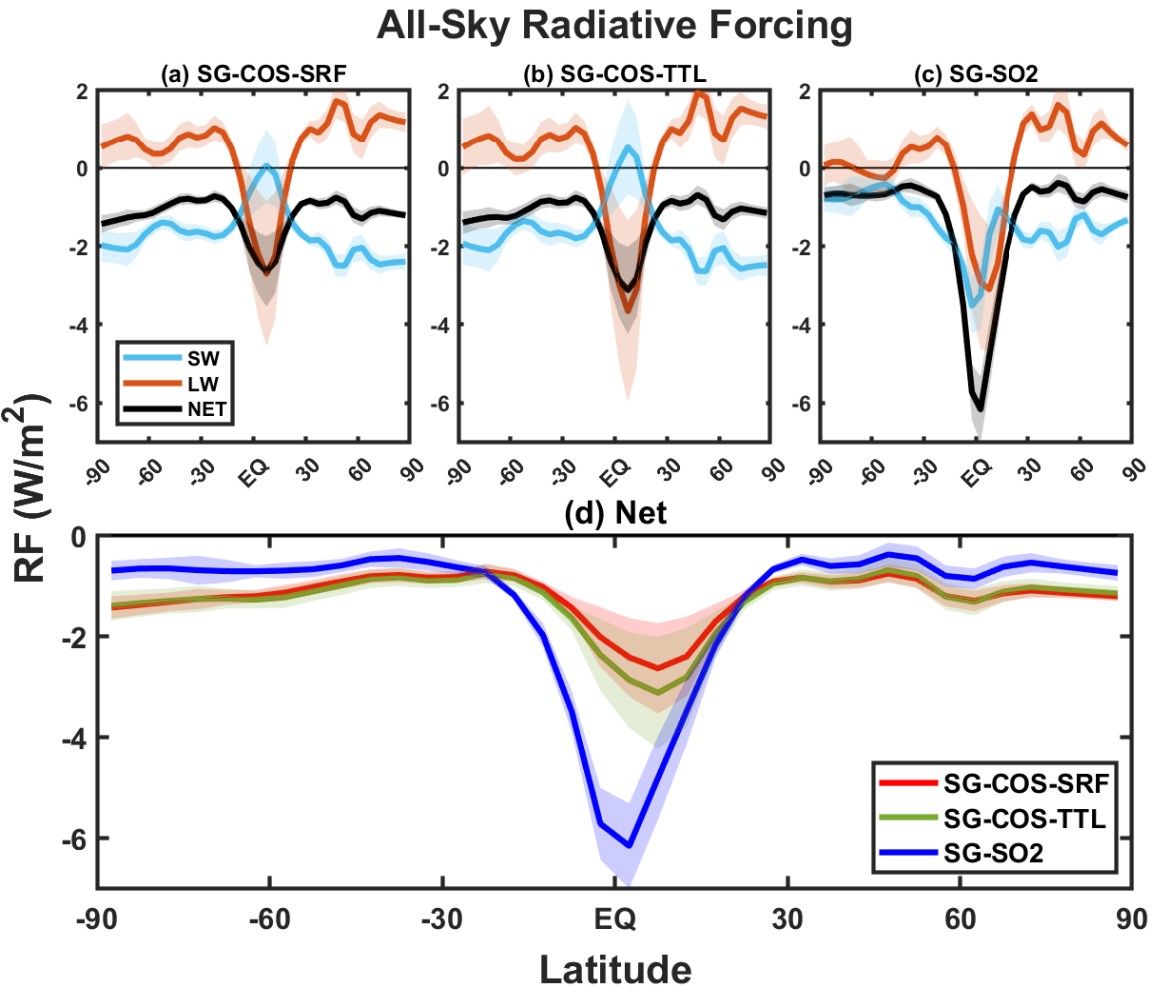

**Figure 9.** a-c) Mean zonal shortwave (cyan), longwave (orange), and net (black) All-Sky radiative forcing (in W/m$^2$) in SG-COS-SRF, SG-COS-TTL and SG-SO2 respectively. d) Comparison of the net radiative forcings from SG-COS-SRF (red), SG-COS-TTL (green) and SG-SO2 (blue). All quantities are annually averaged over the years 2046-2055. Shadings in all panels represent 1 standard deviation in the interannual variability.

## 6 Technical feasibility of SG through COS emissions

We briefly discuss here the technical feasibility of the approach described in this paper, as mainly related to the increase of surface COS emissions (for $SO_2$ injections, see for instance Smith and Wagner, 2018; Smith et al., 2020).

Patent number 3,409,399 (1968) has developed a method for high yield synthesis of COS (93.2-96.6%):

$$CO_2 + CS_2 \xrightarrow{\quad 100-600°C \quad} 2COS$$

$CO_2$ is abundant even in concentrated (90%+) streams, from various natural and industrial sources, particularly with co-operation from states or industries. For example, capturing flue gas from coal-fired power plants is an established technology and may yield over 90% $CO_2$ (Wang et al., 2013). $CS_2$ is produced via numerous means, perhaps the easiest being from coke (carbon) and molten sulfur:

$$C + 2S \xrightarrow{\quad \text{high temperature} \quad} CS_2$$

Approximately 1 million tons of $CS_2$ is produced per year (Madon and Strickland-Constable, 1958), with China consuming approximately half of the global production of $CS_2$ for rayon manufacturing. $CS_2$ is highly unstable and is flammable in air. It is also toxic at low concentrations (10 ppm).

Given the reactions above, about 0.5 Tg of S will produce 0.94 Tg of COS: this amounts to 0.16 Tg of C (coke) and 0.55 Tg of molten sulfur. In the last decade, approximately 70 Tg of sulfur were produced worldwide, so this would constitute an increase in S production of 0.8%. The price varied between $50 and $200 per ton, leading to an annual cost of approximately $25-100 million. Worldwide production of coke was around 640 Tg, so this increase in production is negligible. The price of coke varies between $50 and $100 per ton, leading to an annual cost of approximately $8-16 million. To this we would have to add the cost of $CO_2$, as well as the production and energy costs. Considering an estimate of $400 million per year for each Tg of S between $CO_2$ and production and energy cost, and assuming an effort shared between 1000 locations, this would add up to $400.000 per location per year per each Tg of S. The overall cost is roughly of the same order of magnitude as that in Smith and Wagner (2018) for a stratospheric aerosol deployment at $\sim 20$ km of injection (so different from the injection set-up in our study for SG-SO2), but without the need to develop a new aircraft-based delivery system. For the SG-COS-TTL case, the overall cost would be a combination of the production costs of COS as described above (but almost 10 times less per year to obtain the same AOD as SG-COS-SRF) and those of a deployment in the upper troposphere, which may result to be less expensive than a deployment in the lower stratosphere as needed for $SO_2$.

## 7 Conclusions

We have presented here the results of a modeling experiment with the aim of producing an optically thick cloud of sulfate aerosols in the stratosphere without the injection of sulfate precursors directly in the stratosphere, but rather using increased surface or upper tropospheric emissions of carbonyl sulfide (COS). The low reactivity of COS in the troposphere, where it is not reactive and where it is predominantly absorbed by some soils and by plants, allows for a large portion of its emissions to reach the stratosphere, where it is turned into sulfate aerosols by photo-dissociation and oxidation.

We compare the results obtained in the following injection scenarios: i) 40 Tg-S/yr of COS injected from the surface (roughly 400 times more than the background emissions); ii) 6 Tg-S/yr of COS injected in the equatorial upper troposphere (15 km); iii) 4 Tg-S/yr of $SO_2$ injected in the equatorial stratosphere as prescribed in previous experiments (Kravitz et al., 2011; Visioni et al., 2017a). All experiments result in a similar global optical depth from the produced stratospheric aerosols ($\sim 0.08$), but with different latitudinal distributions: for $SO_2$, as previously observed in various modeling experiments, equatorial injections

result in an increased concentration of aerosols in the tropical stratosphere that tends to overcool the tropics and undercool the high latitudes (Kravitz et al., 2018; Jiang et al., 2019), while also reducing the efficacy of the back-scattering from the aerosols due to the increased size of the particles (Visioni et al., 2018c). On the other hand, with COS emissions, independently from the injection height, the uniform mixing of the gas allows for a more uniform distribution of the produced aerosols in the stratosphere, resulting in increased optical depth also at very high latitude.

The differences in distribution and size of the particles result in different changes to the composition of the atmosphere: smaller particles absorb and heat the stratosphere less, thus resulting in fewer dynamical changes. From a chemical perspective, stratospheric ozone would be impacted differently from the two geoengineering schemes. For $SO_2$ injections, previous studies have shown that the overall effect is the result of a combination of various dynamical and chemical factors that behave differently depending on the latitude and altitude of the aerosols. At low latitudes the increase in lower stratospheric water vapor

produced by the warming of the tropopause layer enhances the halogen-driven destruction of ozone in the lower stratosphere (Tilmes et al., 2018b) due to $NO_x$ depletion. This effect is balanced by reduced ozone destruction in the middle stratosphere due to the slowing down of the $NO_x$ cycle produced by enhanced heterogeneous chemistry (Pitari et al., 2014; Richter et al., 2017; Franke et al., 2021).

Overall, in the case of COS emissions the further increase in surface area density produced by smaller particles increases the

375 inhibition of the ozone cycles in the middle stratosphere, resulting in a net increase in stratospheric ozone and thus in a larger decrease of UV radiation at the surface. Similarly, the larger sulfate burden at high latitudes produces further ozone recovery and thus less UV radiation also at the poles for the COS case.

Our results point to the feasibility of increased emissions of COS as a possible substitute to stratospheric $SO_2$ (or other sulfate precursors) injections to produce stratospheric sulfate aerosols. Surface emissions would sidestep the problem of de-

380 ploying methods not already available to bring the sulfate at those altitudes, including development of novel aircraft (Bingaman et al., 2020). Since COS is already a byproduct of human activities, it might be possible to devise methods of mass-production of the required quantities that may be cheaper than the known proposed methods (Smith et al., 2020). However, this strategy

necessitates a larger amount of emissions to achieve the same global stratospheric AOD, resulting in larger amounts of deposition. Furthermore, while the toxic levels of COS concentrations are orders of magnitude larger than the one achieved in

our simulation (Kilburn and Warshaw, 1995; Bartholomaeus and Haritos, 2006), the effects of prolonged exposure to lower concentrations would have to be assessed; the effect of increased COS concentrations on ecosystems would also require careful investigation. Estimations of the tropospheric radiative effect would also need to be refined to make sure that it is not larger than previously estimated, reducing the efficacy of the aerosol-induced cooling. We have shown that tropospheric injections of lower quantities of COS would produce the same optical depth and indirect effects while resulting in an increase in tropo-

spheric COS concentrations 10 times lower than with surface emissions. This would however still require the deployment of an aircraft fleet as in $SO_2$ emissions, but the technical challenges of reaching 15 km might be less than those faced when reaching 20 km Smith et al. (2020).

Overall, there may be other weak points in geoengineering strategies using COS emissions compared to $SO_2$ that need to be addressed. They would be less easily scalable, and both deployment and phase-out, as we have shown, would require a longer

time-frame compared to the almost instantaneous effect produced by $SO_2$ injections. Considering the dangers to ecosystems presented by a too fast deployment or termination of sulfate geoengineering (Trisos et al., 2018), this might not actually be a large drawback, but it does remove the possibility of rapidly "regulating" the necessary amount of stratospheric sulfate in case of changes in strategy or external conditions (such as a Pinatubo-like volcanic eruption; Laakso et al., 2016). The comparison between our two COS experiments suggest that the mixing happening in the troposphere would not allow any control in the

latitudinal or seasonal distribution of the resulting aerosols, as proposed elsewhere for $SO_2$ injections (MacMartin et al., 2017; Dai et al., 2018; Visioni et al., 2019); however, future investigations may expand on this work by exploring if a different combination of injection altitudes and locations may offer at least some control over the aerosol cloud.

Clearly, this study is intended to be just a pilot study of this method, and further simulations with other climate models, possibly with a coupled ocean and interactive land model to determine the full surface response, are needed. The agreement

between the baseline results presented here and the information present in the literature point to a robustness of our results, but further studies are required to understand different aspects of the climate response: for instance, studies would need to investigate the possible response of vegetation and soils to the increased concentration of COS in the troposphere, and if the efficacy of the sinks would change due to shifts in temperature and precipitation produced by both climate change and the intervention.

Overall, however, the results obtained in this work show that, as a geoengineering technique, emissions of carbonyl sulfide should be further studied and considered by the scientific community as a possible alternative to the others already studied in the literature.

*Data availability.* Data used in this work will be made available through the Cornell eCommons platform before publication.

*Author contributions.* DV and GP devised the study. IQ ran the simulations, analysed the results, produced the figures and wrote the manuscript with the assistance of DV. BK wrote Section 6 and contributed to the final draft of the manuscript.

*Competing interests.* The authors declare no competing interests

*Acknowledgements.* Support for B.K. was provided in part by the National Science Foundation through agreement CBET-1931641, the Indiana University Environmental Resilience Institute, and the *Prepared for Environmental Change* Grand Challenge initiative. Support for D.V. was provided by the Atkinson Center for a Sustainable Future at Cornell University.

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
