# Peer review of "An approach to sulfate geoengineering with surface emissions of carbonyl sulfide"

_Atmospheric Chemistry and Physics, 2021_

## Author Comment (AC1)

**Reviewers' comments are in bold.** Authors' responses are in blue.

**Overview:  This paper proposes using carbonyl sulfide (COS) released at the surface for increasing stratospheric sulfate aerosol.  This proposal has the advantage of not requiring distribution by stratospheric aircraft.  The paper gives a great amount of detail on the differences between the three scenarios examined, a background, a geoengineering scenario using SO2 and one using COS.   It is generally well written.  I recommend publication after addressing the comments below, and most importantly, comment #1.**

We thank the reviewer for their supportive comments. We have responded below to each point.

**1)   Crutzen (2006) https://link.springer.com/article/10.1007/s10584-006-9101-y states "An alternative may be to release a S-containing gas at the earth's surface, or better from balloons, in the tropical stratosphere. A gas one might think of is COS, which may be the main source of the stratospheric sulfate layer during low activity volcanic periods (Crutzen, 1976), although this is debated (Chin and Davis, 1993). However, about 75% of the COS emitted will be taken up by plants, with unknown long-term ecological consequences, 22% is removed by reaction with OH, mostly in the troposphere, and only 5% reaches the stratosphere to produce SO2 and sulfate particles (Chin and Davis, 1993). Consequently, releasing COS at the ground is not recommended."**

**Based on this, first,  this paper should note that this has been proposed before (and therefore may not be so novel) , and discounted due to potential ecological damage.  There is discussion at the end of the paper that this sort of thing should be looked at.  That discussion should be at the start of the paper.**

We expanded the discussion about ecological impacts following the suggestions of point 1) and we added concerns in the introduction about the unknown effects on the efficiency of the uptake by plants and soils in case of higher concentrations of COS, as also suggested by the comment of Dr. Whelan. We added the following phrases:

*"Not much is known however about the response of ecosystems in the presence of high concentrations of COS: Stimler et al. (2010) showed that high levels of COS enhance the stomatal conductance of some plants, which might in turn have other unforeseen effects; further, Conrad and Meuser (2000) proposed that high COS concentrations may interact with soils and possibly change soil pH"*

Further, following comment 6) from the reviewer, we also analyzed a new set of simulations with injections of COS right below the tropopause, that might allow to bypass part of the concerns regarding too high surface COS concentrations. The manuscript has been modified in multiple places to include the new results.

**Second, is the balance between global warming due to COS vs cooling due to aerosols produced taken into account in this study?  Bruhl et al. 2012 state**

**"Further, using a chemical radiative convective model and recent spectra, we compute that the direct radiative forcing efficiency by 1 kg of COS is 724 times that of 1 kg CO2. Considering an anthropogenic fraction of 30 % (derived from ice core data), this translates into an overall direct radiative forcing by COS of 0.003 W m−2 . The direct global warming potentials of COS over time horizons of 20 and 100 yr are GWP(20 yr) = 97 and GWP(100 yr) = 27, respectively (by mass). Furthermore, stratospheric aerosol particles produced by the photolysis of COS (chemical feedback) contribute to a negative direct solar radiative forcing, which in the CCM amounts to −0.007 W m−2 at the top of the atmosphere for the anthropogenic fraction, more than two times the direct warming forcing of COS. Considering that the lifetime of COS is twice that of stratospheric aerosols the warming and cooling tendencies approximately cancel." They also say "Therefore, if we account for indirect chemical effects in GWP calculations, also customary for gases such as methane (IPCC, 2007), it follows that COS has almost no net climate impact." So, if the warming and cooling tendencies cancel, is there an actual advantage to increasing COS emissions? At line 280 this paper states that the COS case produces an RF of .17 W/m2. What is the RF due to the increased sulfate aerosol produced by the COS? How do these results compare to the conclusions of Bruhl et al. 2012?**

In the revised version, we have tried to clarify this point.

We estimate the RF due to the enhanced COS based on the definition of GWP as in Bruhl et al. (2012) and the result is a global warming uniformly distributed latitudinally of 0.17 $W/m^2$.

Under clear sky conditions, we calculated a net clear sky forcing of -2.01 $W/m^2$ due to scattering of solar radiation by sulfate particles (and absorption of LW). This is almost uniformly distributed latitudinally (fig. S6 in the original supplementary) as it follows the distribution of the aerosols and their effective radius (r_eff). The value of stratospheric r_eff determines the efficiency of the interaction of sulfate aerosols with the solar radiation, which peaks at a value of 0.5 µm. In SG-COS the stratospheric r_eff has a global value of 0.46 µm (compared to 0.1 to 0.2 µm in the unperturbed atmosphere in Bruhl et al. 2012 and to 0.52 in SG-SO2) and it is uniformly distributed at all latitudes.

When the contribution of background clouds and cirrus thinning is included, the overall net forcing is -1.52 $W/m^2$ that is more than 2 times greater than the direct warming forcing of COS in the background condition in Bruhl et al. 2012.

We have revised the manuscript to say: "*Overall, this results in a radiative forcing from the COS increase of 0.17 $W/m^2$.*

*The main contributions of sulfate aerosols and clouds are summarised in tables S5 and S6 for SG-COS and SG-SO2, respectively. The contribution of sulfate aerosol is the sum of the cooling effects given by the efficient scattering of solar radiation by particles of radius of 0.5 µm and the absorption of LW by larger ones.*

*Globally, the estimated values are similar for the Clear-Sky SW and LW forcing from the sulfate aerosols: in terms of the latitudinal distribution, however, SG-SO2 presents a peak in the tropics whereas the forcing from SG-COS is much more latitudinally even.*

*The reduction in optical depth from cirrus clouds (see table 1) produced by the aerosols (Kuebbeler et al., 2012; Visioni et al., 2018a) results in a net negative radiative forcing.*

*This is given by the balance between the positive RF in the shortwave (SW) due to the reduction of reflected solar radiation and the negative RF in the longwave (LW) due to the decrease of trapped planetary radiation, which reduces the contribution to the greenhouse effect. "*

**2) The text makes the statement "it is not a toxic gas for humans or ecosystems: negative effects have been found only when concentration exceeds 50 ppm (100,000 times more than the background mixing ratio (Kilburn and Warshaw, 1995; Bartholomaeus and Haritos, 2006)." The Bartholomaeus and Haritos paper does not have that statement, and the Kilburn and Warshaw paper is about H2S and not COS. Different references are needed to say that there are no negative effects on plants and ecosystems.**

The reviewer is correct on both counts. In the revised version, firstly we have removed the mention of ecosystems and focused on human health. For the latter, we have removed the reference to Kilburn and Warshaw, and added the reference to Svoronos and Bruno, 2002, which is where the value of 50 ppm was from (and that is later cited by Bartholomaeus and Haritos, 2006). In the 2002 work, the authors state "Chronic exposure of 50 ppm (mass/mass) carbonyl sulfide (to noncholesterol fed rabbits) of between 0.5 and 12 weeks showed no histotoxic effect on the intimal or subintimal morphology of coronary arteries or the aorta. Similar studies of 50 ppm (mass/mass) carbonyl sulfide exposure to rabbits for 7 weeks showed no effect on the myocardial ultrastructure, and only a slight elevation of the mean serum or aortic cholesterol concentration was observed."

We have revised the manuscript to say: "*In the concentrations found in the atmosphere, it is not a toxic gas for humans: negative effects have not been found even at around 50 ppm, which is 100,000 times more than the background mixing ratio, and for long exposure times in mice and rabbits (Svoronos and Bruno, 2002). Higher concentrations than that can, however, be harmful (Bartholomaeus and Haritos, 2006)*"

**3) The Budyko, 1978 reference link (Budyko, M. I.: The Climate of the Future, American Geophysical Union, https://doi.org/10.1002/9781118665251.ch7, 1978.) is broken (it says the doi cannot be found, this is apparently an error on the AGU web page.). I believe the publication date is actually 1977, and you can get to the full book at https://agupubs.onlinelibrary.wiley.com/doi/book/10.1029/SP010. The reference should be to the full book, since chapter 7 doesn't address the point being made. And, in support of the text (actually proposing SRM) NAS 1992 (National Academy of Sciences (NAS): 1992, Policy Implications of Greenhouse Warming: Mitigation, Adaptation, and the Science Base, Panel on Policy**

**Implications of Greenhouse Warming, Committee on Science, Engineering, and Public Policy, National Academy Press, Washington DC, 918 pp, https://www.nap.edu/catalog/1605/policy-implications-of-greenhouse-warming-mitigation-adaptation-and-the-science should be added.**

We thank the reviewer for catching the broken reference. We have updated it as suggested and added the one to the 1992 NAS report.

**4) Line 72: It should be noted at the beginning of the paper how the increase of 40 Tg-S/yr compares to current emissions. I finally found the factor of 400 in the conclusions.**

We added the information regarding the emission of COS in background condition and the reference to the table in the Supplementary Material where the globally-annually averaged COS sources and sinks are listed for BG and SG-COS.

*"The first geoengineering experiment, SG-COS, tries to produce a significant stratospheric aerosol burden by enhancing current anthropogenic surface emission sources of COS (0.12 Tg-S/yr, see table S1) by up to 40 Tg-S/yr."*

**5) Line 105-106 says "This means an increase of 0.8 ppbv with respect to background condition, that would produce a direct RF negligible if compared to other well mixed greenhouse gases." First, change to "a direct RF that is negligible compared to other well mixed greenhouse gases." Second, is the RF negligible to the negative forcing caused by the increased stratospheric aerosols?**

We have changed the phrase as suggested. In 2075 there is still an increased OD from the stratospheric aerosols, so the direct RF from the GHG is still smaller than that from the aerosols. We have added this to the previous phrase.

**6) And, another experiment that could be run. Instead of emitting a large amount of OCS at the surface, what about emitting at the tropical tropopause? It would require significantly less material, and may produce similar results, with a higher aerosol layer and similar latitudinal distribution. It would also avoid any issues with ecological damage due to increasing surface COS amounts.**

We thank the reviewer for this suggestion. We ran a new set of experiments injecting 6 Tg-S of COS at 16 km of altitude and at the equator (0ºN) to test such an hypothesis. We found that the same stratospheric burden of sulfate can be achieved with substantially less injection of COS (and much less increase in tropospheric COS). The manuscript has been substantially changed in places to reflect the new results.

---

## Author Comment (AC2)

**Overview: This study discusses a new potential approach for sulfate geoengineering using an enhancement of COS at the surface. The study is very well organized, clearly written, and nicely presented. The paper should be published as an important contribution to possible approaches for sulfate geoengineering and their effects. I am supporting the publication after the authors considered the following points listed below. While this is a minor revision in terms of workload, I am suggesting major additions to the discussion and the abstract.**

**Here are some major concerns considering this new approach that has not been clearly addressed in the paper:**

**1) COS is more uniformly distributed in the troposphere and has a much longer lifetime than sulfate. This however suggests that there is much less control using COS than for example using SO2 injections for sulfate geoengineering and limits the potential use of a feedback control algorithm to modulate the amount of cooling in different hemispheres. However, earlier studies by Kravitz et al. (2017) have shown the (game-changing) potential of using a feedback control algorithm to reduce side-effects for instance to reach surface temperature targets or other impact-relevant targets. This is a major drawback of this approach in addition to the lack of the rapid regulation of the injections in case of a large volcanic eruption as already pointed out in this paper.**

We have expanded the final discussion to better clarify this point raised by the reviewer.

**2) Due to its toxicity, there is seems to be a hard limit in using COS. This study increased surface COS concentrations to reach 35ppb, which is very close to the possible limits of 50ppb, as stated in the text. The first concern that needs to be at least mentioned for future work is to identify how reliable the studies are that estimate this toxicity considering potential long-term increases of COS. Secondly, the enhancement of COS reduces AOD by 0.08 with a radiative forcing of -1.3W/m$^2$, which may be translated to less than 0.5 degrees of cooling. Due to the uncertainty in different models, the cooling could be less and may not be sufficient. Furthermore, due to the long phase-in time, it will take a long time to find out how much cooling can be achieved. What if more cooling is needed? Would COS be eventually replaced by SO2 injections? What is the point then of using it in the first place, while adding the danger of exposing humans and ecosystems to a toxic pollutant? To me, this is a major issue that may be a showstopper for considering this method.**

Given the feedback received, we have considerably expanded the discussion in places to further consider the problems that would be encountered with this approach, highlighting uncertainties in the response. Our new set of experiment, which consider the injection of COS below the tropopause (6 Tg-S of COS at 16 km of altitude and at the equator (0ºN)), further highlights the possibility of using COS instead of other precursors.

**3) It needs to be more clearly stated that surface UV is largely reduced with this method, and this can be harmful to humans and ecosystems. A more detailed discussion with the region and season would be helpful.**

[Figure]

**Fig. 8**. Zonal UVB and UVA surface changes per each season in percentage with respect to BG case in SG-COS-SRF (panels a and b, respectively) and SG-SO2 (panels c and d). All quantities are averaged over the years 2046-2055.

We have updated Fig. 8 (see above) and the related discussion to further consider seasonal changes and to highlight the impact of UV changes at the surface.

**Detailed comments:**

**Abstract: The last sentence in the abstract stated that COS emissions are feasible. It may be technically feasible, but I think, the authors need to also point to the drawbacks of this approach in the abstract and conclusions, including the limitations compared to stratospheric SO2 injections.**

We have updated the manuscript, also in light of the new set of experiments, to better clarify challenges and limitations with this approach.

**Line 19: I don't think, this type of intervention can be classified as a "short-term" intervention, earlier studies have shown, that even a "Peakshaving" scenario may require injections between 80-160 years (Tilmes et al., 2016, 2020).**

We removed the term "short-term".

**Line 22: "optically active" is somewhat strange. You could maybe say, "the aerosol layer is thickened and therefore reflects more sunlight…"**

We have substituted with "*to obtain a cloud of aerosols capable of reflecting a portion of the incoming sunlight*"

**Line 27: I don't understand what you mean with "any proposed compound would quickly react to form sulfate aerosols", only sulfur will form sulfate aerosols. Other components may be coded with sulfates that are in the atmosphere. Is that what you mean?**

Any compounds referred to the sulfur species that could be injected, including the largely discussed $SO_2$ injection and the injection of $H_2SO_4$ vapor (Pierce et al., 2010). COS falls in the same category.

**Line 38: Is this the tropospheric lifetime? What is the stratospheric lifetime.**

35 years is the tropospheric lifetime which includes only photochemical reactions (no land sinks). The stratospheric lifetime is about 10 years (due to an increase in photolysis). If land sinks are included, the overall lifetime is 3.8 years. We tried to clarify this in the manuscript.

**Line 65: Since you are looking at UV, please also state what photolysis scheme is used in the model and if photolysis varies with aerosol concentrations.**

As we detail in Section 4, we used the most recent version of the TUV code to determine UV changes at the surface.

**Line 72: Is there a reason why increase COS emissions are placed at the same locations as the anthropogenic emission?**

We placed the increased COS emissions according to the distribution of industrial sources of $CO_2$ and $CS_2$ , as those would be the likely sources used to eventually produce the larger fluxes of COS.

**Line 101: Could you add an estimate of how much surface cooling one expects with an increase in 0.08 AOD? I would be probably less than half a degree of cooling considering the related GeoMIP experiments, is that correct?**

We added a reference to Visioni et al. (2021), where such an evaluation was performed for G6 models. We added the following phrase: "*In the GeoMIP G6sulfur experiment (Visioni et al., 2021b), the average global surface cooling reported by 6 Earth system models for a similar stratospheric OD was 0.46 K.*"

**Line 104: Didn't you state that the lifetime is 35 years, not 3.8 years? What is different here?**

The purely tropospheric **chemical** lifetime of COS is 35 years and decreases to 3.8 years when including both chemical reactions and dry deposition.

**Line 106-107. I don't follow, what is meant here. 0.8ppbv of what, and why do you refer to the RF of other greenhouse gases? Do you mean, that after 20years, COS values have declined close enough to the background to not have a significant impact on the RF?**

0.8 ppbv refers to the tropospheric concentration of COS. We want to emphasize the role as a greenhouse gas in the decreasing phase when sulfate aerosols are still in the stratosphere reflecting solar radiation. By comparing the radiative forcing of COS with that of other greenhouse gases that we have included in the RF discussion, we want to say that COS does not produce a significant impact on RF.

We changed "This means an increase of 0.8 ppbv with respect to background condition, that would produce a direct RF negligible if compared to other well mixed greenhouse gases." to:

"*This means an increase of COS of 0.8 ppbv with respect to background condition, that would produce a direct RF negligible.*"

**Line 230: The catalytic NOx cycle is decreased with more surface area density, which results in less ozone loss. Isn't the inhibition of denitrification more important in high latitudes and for cold temperatures, and less in other latitudes**

As shown in Tilmes et al. (2017), NOx cycle changes are far more important at low latitudes but we've tried to clarify this in the revised manuscript.

**Line 234: "its photodissociation" what is "its" referred to here?**

"Its photodissociation" refers to the ozone. We changed the sentence to make it clearer like this:

"*Increasing stratospheric ozone affects UVB at the surface because it is absorbed by ozone during its photodissociation.*"

**Line 240: change "UVA decrease is everywhere negative in both SG experiments" to "UVA decreases everywhere in both SG experiments"**

Corrected.

**Table5 and Figure 8: You are nicely showing total column ozone variations with region and seasons, and it is very clear that there are differences in sign in the response. Illustrating UV changes annually is not a very meaningful measure especially for high latitudes. I would strongly recommend expanding this figure and showing 4 seasons instead or in addition to the annual values. I would also expand the discussion on the UV impacts, and why UV-A and UV-B and shown separately. Not sure if there is any reference to Figure 8 in the text?**

This is discussed in conjunction with item (3). We have amended the text accordingly.

**Line 269: reference should be Bruhl et al. (2012)**

Apologies. We have fixed the name in the paper and references.

**Line 289: should be (Figures 9 and S6)**

Corrected.

**References**

Pierce, J. R., Weisenstein, D. K., Heckendorn, P., Peter, T., and Keith, D. W. (2010), Efficient formation of stratospheric aerosol for climate engineering by emission of condensible vapor from aircraft, Geophys. Res. Lett., 37, L18805, doi:10.1029/2010GL043975.

---

## Author Response (AR2)

**Reviewers' comments are in bold.** Authors' responses are in blue.

The authors have addressed my comments appropriately and it was an interesting idea to add another experiment to the study. I am fine with the publication of this paper if the following minor comments have been addressed.

**Minor comment:**

**Abstract: One more sentence at the end of the abstract would help to point to the drawbacks of this approach including, the need for research in particular on the impact on ecosystems, and regional climate impacts.**

We added a new sentence at the end of the abstract to summarize pros and cons of the method:

*"However, our assumption that the rate of COS uptake by soils and plants does not vary with increasing COS concentrations will need to be investigated in future works, and more studies are needed on the prolonged exposure effects to higher COS values in humans and ecosystems."*

**Figure 9, bottom panel legend seems to be wrong and not aligned with the Figure caption.**
Corrected.

**Figure 8: bottom panels all both labeled b) but should be c) and d)**
Corrected.

**Line 217: "for the SG-COS experiment". Please clarify here that you are only performing comparisons with the surface injection case.**

We included the clarification, and further specified below (see next comment) why we're only comparing against one of the two experiments.

**Line 218: is not clear what "SG-COS-TTL is equivalent SG-COS-SRF" means? It would be helpful to explain that since SAD is almost the same in those two scenarios, SG-COS-TTL is expected to show very similar results, or so.**

"Figure 7 shows the ozone changes in SG-COS-SRF and SG-SO2 with respect to the BG case (SG-COS-TTL is equivalent to SG-COS-SRF)."
Changed in:
"Figure 7 shows the ozone changes in SG-COS-SRF and SG-SO2 with respect to the BG case. As expected from the similar value and distribution of the SAD, in SG-COS-TTL ozone changes are equivalent to SG-COS-SRF (and are therefore not shown)."

**Line 267: comma before "as well as", also it seems like the COS experiments do not show an increase for SON over Antarctica. Also, in DJF there is some increase in UVB in the Northern Hemisphere. In general, there could be a little more discussion on the seasonality of UV.**

"In all SG experiments, the negative changes of UVB radiation at surface, except in the Antarctic region, are related to the variation in stratospheric ozone, as well as the interannual variation that increases towards the poles, due to the seasonal variation of ozone, as discussed before. "

We changed it to:
"In all SG experiments, the negative changes of UVB radiation at the surface, except in the Antarctic region, are related to changes in stratospheric ozone, as well as the interannual variations that are larger at the poles, due to the seasonal variability, as discussed before. In the Antarctic Spring (SON) the ozone depletion is enhanced in SG-SO2 while in SG-COS-SRF it is limited to the month of October, with differences compared to BG of less than -5 DU. Therefore, the UVB change compared to BG for SON over Antarctica remains negative in SG-COS-SRF with a value of -2.7% versus a +5.8% increase in the SG-SO2 experiment. In DJF, on the other hand, a small increase of UVB is observable at mid to high latitudes in the Northern Hemisphere. This is connected to an observable decrease of stratospheric ozone in the same locations, possibly due to a reduced advection of air from the tropics."

**Line 312: micrometer?**
Corrected.